# Supporting a Resilience Observatory to Climate Risks in French Polynesia: From Valorization of Preexisting Data to Low-Cost Data Acquisition

**Jérémy Jessin** [1,2], **Charlotte Heinzlef** [3], **Nathalie Long** [2] and **Damien Serre** [1,4,5,*]

1    IFREMER, ILM, IRD, UNIV. POLYNESIE FRANCAISE, EIO UMR 241, BP 6570, Tahiti 98702, French Polynesia; jeremy.jessin@doctorant.upf.pf
2    UMR LIENSs, La Rochelle Université—CNRS, 2 rue Olympe de Gouges, 17000 La Rochelle, France; nathalie.long@univ-lr.fr
3    CEARC, Université Paris Saclay, UVSQ 11 boulevard d'Alembert, 78280 Guyancourt, France; charlotte.heinzlef@uvsq.fr
4    UMR 7300 ESPACE—Avignon Université, 74 rue Louis Pasteur—Case 19, 84029 Avignon, France
5    ARIACTION, Ecole d'Urbanisme et d'Architecture du Paysage, Université de Montréal, Montréal, QC H3T 1J4, Canada
*    Correspondence: damien.serre@univ-avignon.fr

**Abstract:** Climate change has an ever-increasing impact on island territories. Whether it is due to rising sea levels or the increase in recurrence and intensity of extreme events, island territories are increasingly vulnerable. These impacts are expected to affect marine and terrestrial biodiversity, human occupation (infrastructure) and other activities such as agriculture and tourism, the two economic pillars of French Polynesia. While the current and future impacts of climate change on island territories are generally accepted, data acquisition, modeling, and projections of climate change are more complex to obtain and limitedly cover the island territories of the Pacific region. This article aims to develop methodologies for the acquisition and exploitation of data on current and future climate risks and their impacts in French Polynesia. This work of acquisition and valorization is part of a research project for the development of an observatory of resilience to climate risks in the perspective of building a spatial decision support system.

**Keywords:** climate change; climate data; aerial UAV data acquisition; climate projections; spatial support process; resilience observatory; pacific islands; French Polynesia



## 1. Introduction

Island territories are some of the most vulnerable areas to climate variations. In most of these territories, the inhabitants, infrastructure, agriculture, recreational activities and/or tourism are concentrated in coastal areas, thus developing a linear urbanism extremely vulnerable to climatic variations and rising sea levels [1]. Among the territories most impacted by climate change with the highest rates of risk-related losses of GDP, two-thirds are island territories with an annual average loss of between 1 and 9% GDP [2]. The Asian-Pacific islands (Figure 1) [3] are considered the most highly exposed to natural disasters in the world, with the highest disaster-related deaths, representing 75% of the global mortality between 1970–2011 [4,5].

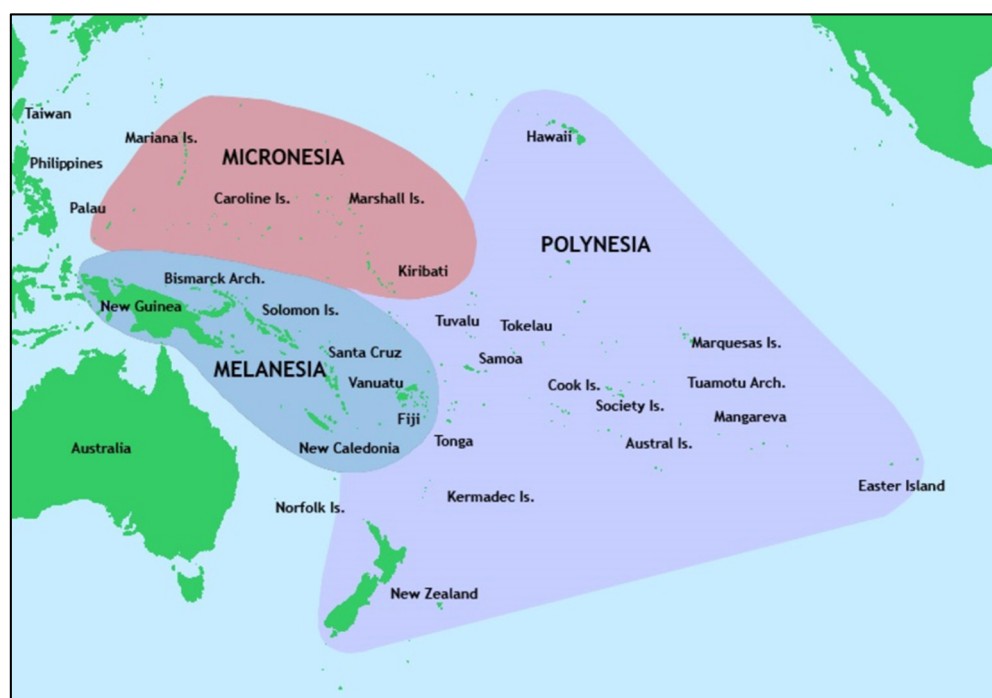

**Figure 1.** Main islands in Pacific Ocean [6].

The vulnerability of this region is thus proven and attested (Table 1). This vulnerability index is measured with a country's exposure, sensitivity and ability to adapt to the negative impact of climate change. There are six sectors analyzed: food, water, health, ecosystem service, human habitat and infrastructure. In comparison with European countries, the average vulnerability rate in Europe is between 0.2 and 0.3 [7].

**Table 1.** Examples of vulnerability rank of some Pacific islands, adapted from (Kuruppu and Willie, 2015) [8].

| Pacific Islands | Population | Vulnerability Index (GAIN Ranking) |
|---|---|---|
| Samoa | 187,820 | 0.428 (127) |
| Salomon Islands | 561,000 | 0.514 (164) |
| Timor-Leste | 1,066,409 | 0.517 (165) |
| Vanuatu | 264,652 | 0.429 (128) |

Sea level rise is one of the major hazards in this region, as well as floods, tropical hurricanes and tsunamis, which are all increasing in recurrence and intensity due to climate change. These hazards can have a negative impact on the population and produce significant material damage, especially within territories that are exposed and vulnerable [9–11]. Because of their geomorphological specificities, the islands and atolls [11,12] of the Pacific are specifically vulnerable to sea level rise. Over a significant portion of the past century, sea level rise reached an average of between 1.3 mm and 1.7 mm per year on a global scale. Unfortunately, this rise accelerated at the end of the 20th century and has increased to an average of 2.8 mm and 3.6 mm per year [13]. However, this evolution is not identical on a global scale, and the differences observed are even greater in the Pacific region. The pacific averages are four times higher than the global mean, reaching about 12 mm per year [14]. Due to the linear urban planning specific to these islands, the economic, social, political and infrastructural stakes are all concentrated on the coastal fringe, which increases their vulnerability to risks. This situation is not likely to change with regard to the current projections. It is predicted that the sea levels in the Pacific region will not only rise, but rise even faster than everywhere else [15]. In some areas of the Pacific, the sea level is expected

to be 10 cm higher than the current conditions, keeping in mind that it is necessary to take into account the margins of uncertainty which can be important [16,17].

Faced with these hazards, their increasing intensity and recurrence and their catastrophic impacts on human occupation, infrastructure, the economy, the environment, the well-being of populations, etc., risk management has evolved by integrating new concepts. Resilience is one of these concepts needed to implement risk-management strategies that are better adapted to climate change and related uncertainties [18]. However, the resilience concept is still complex to operationalize and integrate at the local level [18]. In addition, existing models have very few long-term perspectives, making resilient risk management strategies more convoluted. In light of the current climate of increasing risks and lack of adequate and long-term operationalization of resilience, a Pacific Risk Resilience Observatory project was developed [19]. While this study will focus on sea level rise on the island of Tahiti (see Section 3) in French Polynesia, there are a variety of other hazards present within the archipelagos. The observatory will serve as a tool to agglomerate data on not only sea level rise but the following main hazards of French Polynesia as well:

- Tsunamis are one of the major hazards in the Pacific Ocean, because of the "Pacific Ring of Fire" [9]. French Polynesia, and especially the Marquesas Archipelago [9,20,21] may be strongly impacted by this hazard. For instance, some tsunami run-ups have reached 10 m in Nuku Hiva (1946) or Hiva Oa (1946) [22].

- Cyclones are the second major coastal hazard in French Polynesia. For instance, in the Austral Archipelago [21] the cyclone frequency is roughly one event every 7 years [23]. Even if the cyclones are mostly contained west of the 150° meridian, because of the El Nino oscillations, they may occur in the Society and Tuamotu archipelagos [21]. A general average of about ten hurricanes per season can be proposed [14]. Although it is difficult to establish a trend, the major projections present an overall stability in the intensity of hurricanes but an increase in their frequency [13,24]. The impacts include stronger winds and rainfall, sea-level rise, coastal inundations, and storm surges, all of which are accompanied by damages related to these risks.

- Finally, the risk of fluvial flooding is increasingly present and has consequences that are all the more important within this territory wedged between a lagoon and a mountain. The 2017 floods in Tahiti are a relevant illustration of this. Within 6 h, 200 mm of water fell, impacting between 800 and 900 homes, i.e., 4000 people affected. The material damage was considerable, destroying critical infrastructure such as bridges connecting the two parts of the island. Unfortunately, the trends show that these floods are increasing in intensity and recurrence due to climate change, with floods occurring again in 2018 and 2020.

The objectives of this article are trifold and will begin by presenting the main objectives of the resilience observatory and the tasks that make it up (Section 2). This spatial decision support system is intended to be fed by a flux of data which is exemplified by the second objective of this article. This second objective analyzes the knowledge and comprehension of hazards and their associated risks with pre-existing data by using the island of Tahiti as a case study to analyze the potential impacts of sea level rise (Section 3). Finally, the article will conclude on the need to acquire new types of data, serving as a progress report on the current applications and future developments of a recently developed aerial data source (UAVs) more in line with the techniques and local practices, knowledge and uses (Section 4).

## 2. A Resilience Observatory to Address the Challenges of Resilience Implementation in French Polynesia

The concept of resilience appeared in the 2000s in the field of risk management [25]. Resilience, a multidisciplinary concept, is defined (in the context of risk management) as the capacity of a system to adapt to disruptions, limit their negative impacts and recover a balance following the shock(s) [26]. The associated notions are the capacities for learning [27–29], adaptation [30], rebound, innovation, flexibility, evolution and anticipa-

tion [18,31]. This concept is fully adapted to the systemic complexity of contemporary territories and dynamics, in addition to becoming an imperative need for regional planning in the face of climate risks [18,32]. However, its application in local policies is still limited due to the complexities in its operationalization [33]. These complexities include a multi-disciplinary origin [34], a multitude of definitions [35] and related concepts [36–38], and resilience is complex to integrate into operational and effective local risk management practices [18,32,39]. Even though these concepts show an undeniable theoretical potential, its operational reality and adequacy to address the local challenges of climate change and associated risks is constantly being questioned.

To overcome this operational limitation, numerous studies [33,40,41], models [42,43] and indicators [26,44–47] have been developed. However, the problem is precisely the multitude of these studies and models, thus accentuating the vagueness surrounding the operationalization of the concept of resilience. Local actors get lost in the existing tools: which model(s) to choose, which tool(s), which approach(es), which definition(s) [19]? Developing a unique decision support tool, bringing together different models, aiming to clarify the concept of resilience and support local stakeholders in their decision making to implement local and resilient risk management strategies [19] is essential. An observatory of risks and resilience is therefore currently being developed in French Polynesia and, as a first case study, Tahiti [19,48,49]. Developing a risk and resilience observatory in French Polynesia allows us to respond to the need for a local tool that is economically relevant, unique, and understandable, and adapted to the local territory. This observatory, which is intended to be a spatial decision support tool, has been developed around several tasks (T) (Figure 2) and steps [18]:

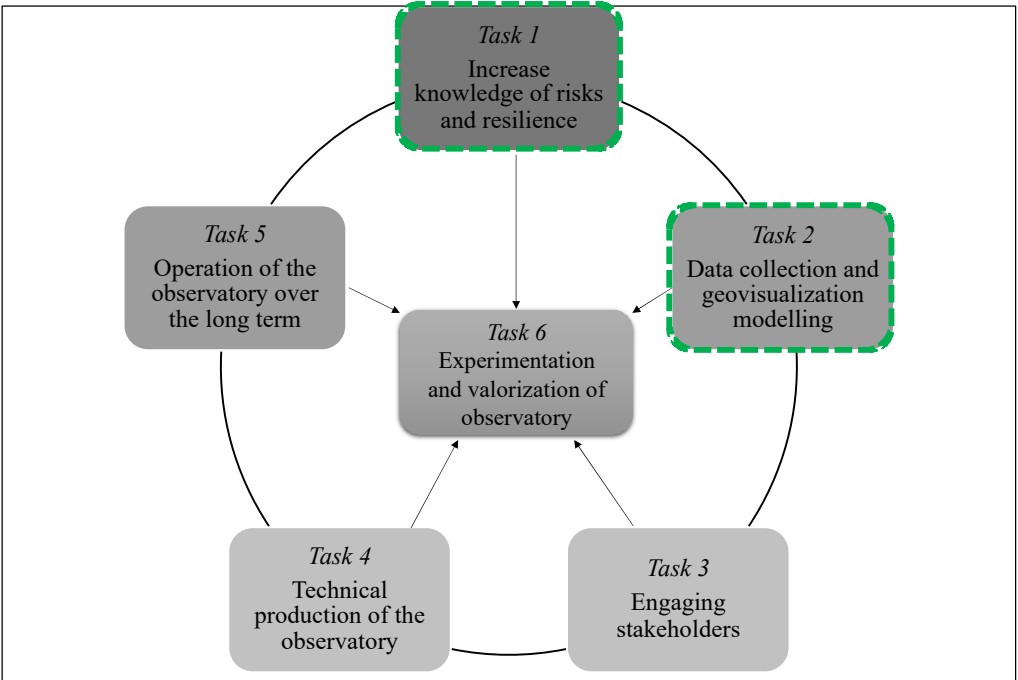

**Figure 2.** Analyzed tasks in this research [19].

T.1 "Increase risk and resilience knowledge": the objectives are to transmit, condense and multiply knowledge on risks, resilience and to identify the linked issues. This step is built around pre-existing local data, processed and valorized in order to produce new knowledge on the territory.

T.2 "Data collection and geo-visualization modeling": this task aims to respond to the limitations of task 1. Pre-existing data may contain limitations, not only in terms of quality, but also in terms of cost, accessibility and repetitive capacity. It is then necessary to

use new means of data acquisition to correspond to local needs and means but also in a long-term perspective.

T.3 "Engaging stakeholders": This task completes T.2 by providing qualitative data. It responds to the future appropriation of the tool by decision-makers, which is a condition for the tool's success. The essential objective is to develop collaborations with local actors and managers.

T.4 "Technical production of the observatory": this task is necessary to gather the results, methods, and data acquired in the previous tasks. The idea is to produce an accessible and functional observatory.

T.5 "Operation of the observatory over the long term": It is necessary to validate and continuously improve the functioning of the observatory in order to perpetuate the observatory in the long term.

T.6 "Experimentation and valorization of observatory": Task 6 is used to validate and finalize the methodology.

This article aims to develop and illustrate the challenges of valorizing existing data on risks (Task 1) but also the necessary acquisition of new types of data through more efficient methods adapted to the territory in question (Task 2). This data-driven research contributes to the need for local risk data and knowledge (Figure 2).

## 3. Impacts and Issues of Climate Risks in Tahiti: Increasing the Knowledge on Sea Level Rise (Observatory Task 1)

French Polynesia is subject to many natural hazards [33]. These phenomena often cause both material and human damage. Tahiti (Figure 3), composed of a main island (Tahiti Nui) and a peninsula (Tahiti Iti), is the largest island (1042 km$^2$) and the most populated of the islands of French Polynesia. The western tropical Pacific, including Tahiti and French Polynesia, is one of the most affected regions in the world today by the rising sea level [50,51]. Most of the studies conducted on sea level rise in French Polynesia are studying the impacts on atoll islands [52–55], but very little research has been produced on high islands such as Tahiti. This recent increase exposes the crucial need to produce data to better understand the future effects of climate change on this island.

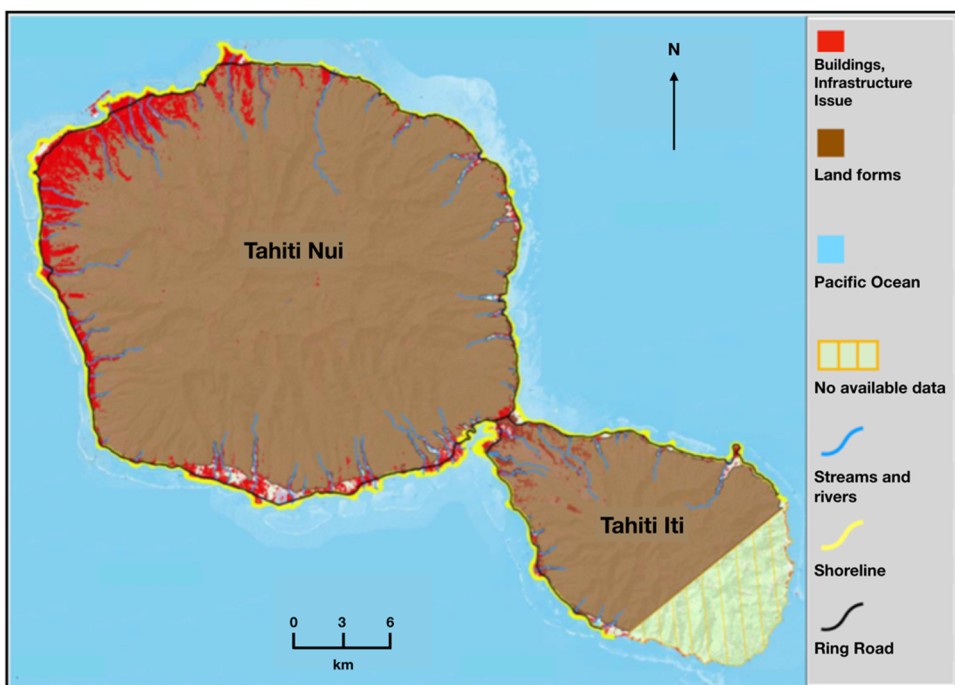

**Figure 3.** Spatial representation of urban issues on Tahiti Iti and Tahiti Nui [56].

Tahiti centralizes the socio-economic, political, urban, agricultural and touristic activities (notably with the only international airport in French Polynesia) of the territory [56]. Papeete is the capital of French Polynesia and the main city of Tahiti, concentrating (in 2017) 70% of the country's total population and 87% of the inhabitants of the Society Islands archipelago according to the "Institut de la Statistique de la Polynésie Française" (ISPF). The specificity of the agglomeration of Papeete lies in its sprawl, which has led to the continuous occupation of the coastal plain in a corridor of a few hundred meters over 60 km long. This has generated a high need for daily travel, a phenomenon reinforced by the concentration of jobs, facilities and services in the city center [57]. The concentration of economic, sociological, political, touristic and urban issues on Tahiti and the Papeete agglomeration makes this area extremely vulnerable to any disruptions. Climate risks are therefore extremely concerning in regard to the impact they can have on the coastal zones [48], impacting not only the territory of Tahiti but also threatening to paralyze the Polynesian territory through a cascading effect [47,58]. In order to be a part of a long-term process, a study integrated to the resilience observatory Task 1 was conducted to model the impact of climate change on sea level rise and related impacts on the Tahitian territory. This analysis was based on pre-existing local data and IPCC forecast scenarios.

As is evident from Figure 3, the majority of the urban development is concentrated in the north-west region of the island. This study will thus focus on the urban agglomeration of Tahiti, which includes the municipalities of Arue, Pirae, Papeete, Faa'a and Punaauia (Figure 4). This geographical zone has a significant economic importance by hosting nearly 58% and 40% of the inhabitants of Tahiti and French Polynesia, respectively (ISPF, 2017). The population since the 1970s has more than doubled in Polynesia and has increased the most in the urban area of Papeete [57]. Its capital, Papeete, is also the political and economic capital of this French overseas country [57]. Faa'a International Airport and the Autonomous Port of Papeete are the only international accesses in the entire territory of French Polynesia. Thus, both economic and migratory links to the rest of the world are located in this urban area, which makes it all the more important for the country since all goods and people must necessarily transit through it before being transferred to other islands [57]. Since 1960, a major wave of urbanization has occurred around Papeete, which has become the center [57]. The high demand for labor in the urban agglomeration and the geographic specificities of Tahiti have led to the continuous occupation of the coastal plain in a corridor of a few hundred meters wide by more than 60 km long [57]. The geography of the island results in a high concentration of population on the coastlines, while the interior of the island is only sporadically occupied [57].

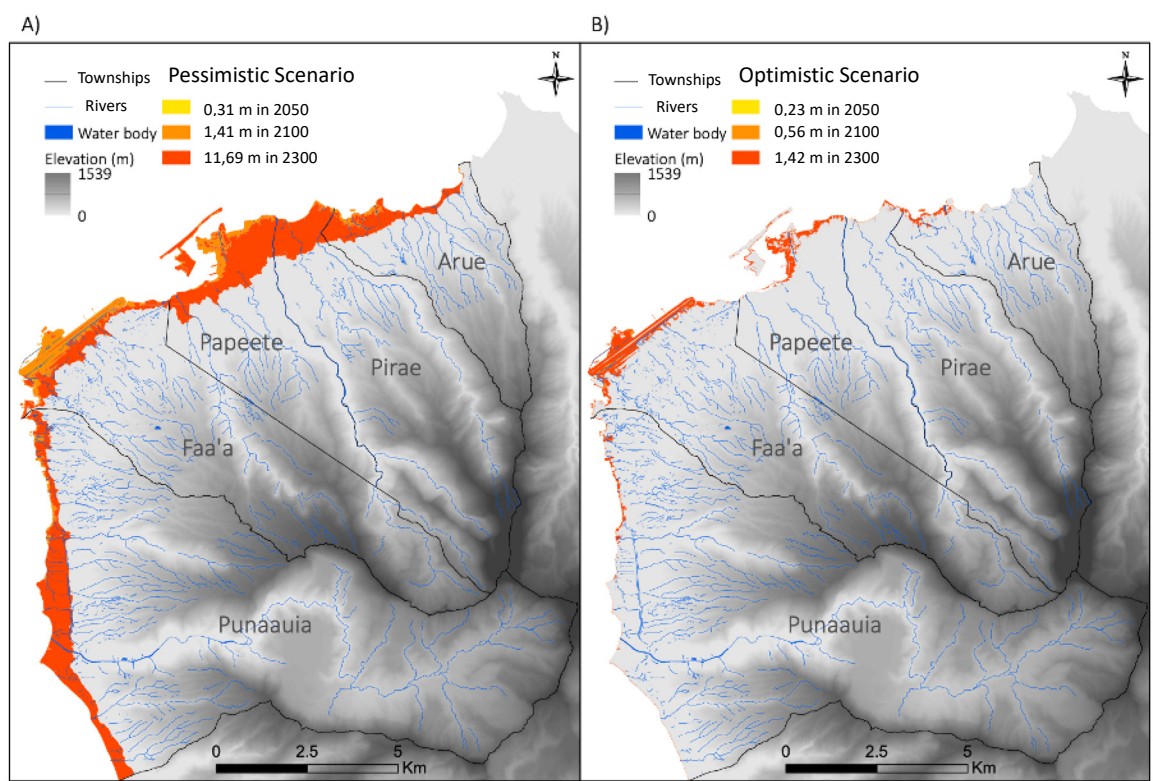

**Figure 4.** Flooded areas in Tahiti. (**A**) Pessimistic scenario flooding impacts from 2050 to 2300 (**B**) Optimistic scenario flooding impacts from 2050 to 2300 [59].

### 3.1. Pre-Existing Data in Tahiti

With increasing pressures on coastal regions, the need to improve our knowledge about these ecosystems is increasing as well. An important step is the capacity to perform repetitive surveys of our environment in order to consistently identify and measure indicators that define the status of our environment [60]. Obtaining elevation data for the creation of 3D models or topographic maps are effective and efficient methods of monitoring these environments capable of informing management decisions [61]. Conducting these surveys comes with its challenges, and methods differ widely based on cost, acquisition time, and accuracy [62]. To obtain said elevation data, aerial approaches dominate the field today, and the most commonly used methods (Table 2) come in the form of aerial photography (by plane/helicopter), satellite images, and light detection and ranging technology (LiDAR).

**Table 2.** Comparison of various available aerial data sources in French Polynesia.

|  | Aerial Photographs (Aircraft) | Satellite Images | LIDAR |
|---|---|---|---|
| Utility | - Natural risk assessment<br>- Urban development evolution | - Vegetation analysis<br>- Shoreline change evaluation<br>- Erosion assessment | - Erosion assessment<br>- Marine submersion<br>- Flood assessment<br>- Reef state mapping |
| Timeline of available data | 1955–2021 | 2003–2017 | 2015 |
| Subject of data | All 5 archipelagos of French Polynesia | All 5 archipelagos of French Polynesia | - West coast of Tahiti<br>- Coast of Moorea<br>- Bora-Bora<br>- Tetiaroa |
| Accessibility | Open source | Available upon purchase | Available upon purchase |
| Benefits | - Rapid acquisition time<br>- Wide temporal availability | - Large spatial coverage<br>- Multi-spectral data | - High spatial resolution<br>- 3D models (DSM, DEM, DTM) |
| Limitations | - Lower spatial resolution<br>- High cost | - Lower precision<br>- Slowest acquisition time | Highest cost |

Aerial photographs and satellite images both have a temporal aspect that is important when observing the evolution of an environment. Some aerial photographs of French Polynesia date back to the 1950s, and this is the only available aerial data at the time. Satellite images, on the other hand, do not date as far back; however, they can provide spectral information that cannot be found in these older aerial photographs. Additionally, the entirety of the islands of French Polynesia can be viewed by satellite images because of their wide capacity for spatial coverage. LiDAR is not as accessible, mostly because of the price of acquisition; however, it has enormous benefits to its utilization, due to its high resolution and capacity for modeling. The LiDAR campaign has been estimated at over EUR 50 million; however, even though its cost of operation limits its repetition and therefore the acquisition of a time series, the high vertical accuracy combined with a resolution of 1 m allows for the production of a reliable DEM that can be used for risk assessment and management in French Polynesia. Despite the high cost, the acquisition of precise aerial data to obtain high-resolution models has been largely dominated by the use of airborne LiDAR and remains unparalleled [63]. Coastal vulnerability to sea-level rise has already been studied on Tahiti Island [64] and could be the subject of further, more precise studies using LiDAR data.

### 3.2. Data and Methods

### 3.2.1. Sea Level Rise Scenarios and Antarctica Factor

To quantify climate change and try to establish how the climate will evolve in the coming years, the Intergovernmental Panel on Climate Change (IPCC) has developed an index that summarizes the different greenhouse gas (GHG) emission scenarios [1]. The RCP (Representative Concentration Pathways) indices [1,65,66] represent a concentration of GHGs that would be contained in the atmosphere according to anthropogenic activities [1,66] such as the type of energy used (fossil or renewable), new technologies developed (e.g., carbon capture), means of transportation, land use (e.g., agriculture, grazing), etc. [1,66]. From there, four standardized models or "pathways" were developed to represent the possible consequences of different anthropogenic climate forces [65,66]. As an example, the RCP 2.6 scenario represents an eventuality where there would be significant global GHG mitigation, lowering the emissions to zero after 2050 [65] (National Climate Change Adaptation Research Facility (NCCARF), n.d.). In this scenario, global average temperatures would increase by only 1 °C (NCCARF, n.d.). However, this scenario is virtually impossible since global temperatures have already reached that threshold [15]. Contrastingly, the RCP 8.5 scenario represents the other extreme, i.e., GHGs would continue to skyrocket by 2100 and continue to increase thereafter. In this situation, temperatures would increase by at least 3.7 °C (NCCARF, n.d.). In terms of sea level rise (Table 3), the RCP 2.6 scenario would rise by 1.42 m by 2300 while the RCP 8.5 scenario would rise by 11.69 m.

The two factors that most contribute to this rise are currently the melting of the ice caps (glaciers and ice sheets) and the thermal expansion of the ocean [67], particularly that of the upper layer (<700 m depth) (IPCC, 2014). To measure changes in sea level, researchers use the "Relative sea level" (RSL) which can be defined as "the height of the sea at a specific location, measured relative to the height of the Earth's solid surface" [65]. The RSL is generally measured locally and the data put together will form a global mean sea level (GMSL) [1,68].

Many uncertainties remain regarding the possible rise in sea level. Indeed, not only does it depend on political decisions related to GHG emissions, but the sea level will also, very soon, depending on the behavior of the Antarctic ice cap [1]. These uncertainties have been raised since the very beginning of global sea level rise modelling [65]. DeConto and Pollard (2016) [69], show that rising global temperatures promote hydraulic fracturing in glaciers. This fracturing releases ice masses into the ocean, accelerating their melting. It is estimated that the melting of Antarctica could exacerbate sea level rise in a disastrous way [70], since it represents the largest ice sheet on Earth today [1]. This is especially problematic when determining the impacts after 2050, given the extremely variable climate

pathways. However, several new methods have been developed to better represent the contributions of the melting of different parts of the continent [1]. One of these methods is to evaluate the response of the ice sheet during the last interglacial periods to project these behaviors with warmer future climates [65]. Thus, the Antarctic ice sheet plays a major role in modelling future sea levels. The SLR rise values used were taken from a study by Kopp et al., 2017 [68]. It was chosen because it takes into account the implication, often neglected, of the Antarctic ice sheet according to the amount of water it retains [65]. The median values of each scenario were used to standardize the analysis between the most optimistic and pessimistic scenarios (Table 3).

**Table 3.** Sea level rise values (Kopp et al., 2017) [68].

| Year | Pessimistic (RCP 8.5) | Optimistic (RCP 2.6) |
|------|-----------------------|----------------------|
| 2050 | 0.31 (m) | 0.23 (m) |
| 2100 | 1.46 (m) | 0.56 (m) |
| 2300 | 11.69 (m) | 1.42 (m) |

### 3.2.2. Data, Methods, and Tools

This study used a Digital Elevation Model (DEM) built from the LiDAR data mentioned previously and processed with a geographic information system. All operations of the data processing were conducted with ESRI's ArcMap. The "Lidar French Polynesia 2015" operation was commissioned by the French Polynesian Urban Planning Department (SAU), led by the Hydrographic and Oceanographic Service of the Navy (SHOM) [71]. These data were acquired by airborne bathymetric and topo-bathymetric LiDARs in May and June of 2015 and covers the north-western coast of Tahiti, the entirety of the coast of the island of Moorea, and the entirety of the island of Bora Bora. This LiDAR DEM is currently used today for the creation of the "Prevention Plan of Risks" (PPR), making it possible to highlight vulnerable areas exposed to natural hazards and to map natural risks (landslide, floods or marine submersion) in order to take them into account for construction and land-use planning.

In order to conduct the analysis of flooded areas on Tahiti with regard to the RCP scenarios, the LiDAR data were transformed into a 1 × 1 m accuracy DEM, which covers from the barrier reef to approximately 1 km inland. This DEM (formed from Lidar imagery points) has an accuracy of 3 to 4 points per square meter [71]. Since the provided DEM did not contain data underneath the buildings, an inverse distance weighting (IDW) interpolation was performed to calculate the value of the empty points. Area losses were then calculated to map sea level rise for the two GHG emission scenarios and the three base years.

In order to conduct analysis of the infrastructure impacted by sea level rise, the LiDAR data was cross-referenced with data on the different categories of buildings, also obtained from the SAU of Papeete. The building categories determined by the SAU of Papeete are: religious, education or training, primary and secondary sector, service (tertiary), residential, miscellaneous and public service buildings. To classify the buildings on the island, the SAU first digitized them in a shapefile format. Then, teams went to the field to determine the category of each building. When a building has more than one function, it is the second category that dominates over the residential category. The main objective of this portion of the study is to evaluate the amount and type of infrastructure affected according to the different sea level rise scenarios. In order to do so, a table of attributes was created between the buildings and the different matrices reclassified according to the different scenarios. Thus, each building was associated with the lowest pixel value it touched. This new table was joined to the vector layer of buildings to determine the type of each building affected. These data were then extracted for the five municipalities studied in order to show the differences in terms of impacts for each of them. These data were then extracted for the five study communities to show the differences in impacts in each community. As this was

carried out, the data for the number of buildings affected in each year and the building types were entered into Excel for figure production.

### 3.3. Results

The results are divided according to the flooded areas and the impacted infrastructures. They are presented according to the SLR projections for 2050, 2100 and 2300 [65].

#### 3.3.1. Simulations for Flooded Areas

2050:

Within the urban area of Tahiti, the flooded land represents only 0.043 km$^2$ (4.3 hectares) or 0.023% of the territory of Tahiti in a pessimistic scenario. Mainly, the beaches such as the one in Arue are affected, as well as the outlets of large rivers (Figure 4). The two most affected municipalities by 2050 will be the municipalities of Faa'a, with a land loss of 0.054%, or 0.019 km$^2$ (1.9 hectares), and of Papeete with a loss of 0.041%, or 0.007 km$^2$ (0.7 hectares) (Figure 5). Arue comes in at distant third with a land loss of 0.027%, while Punaauia and Pirae are at 0.010% and 0.008%, respectively.

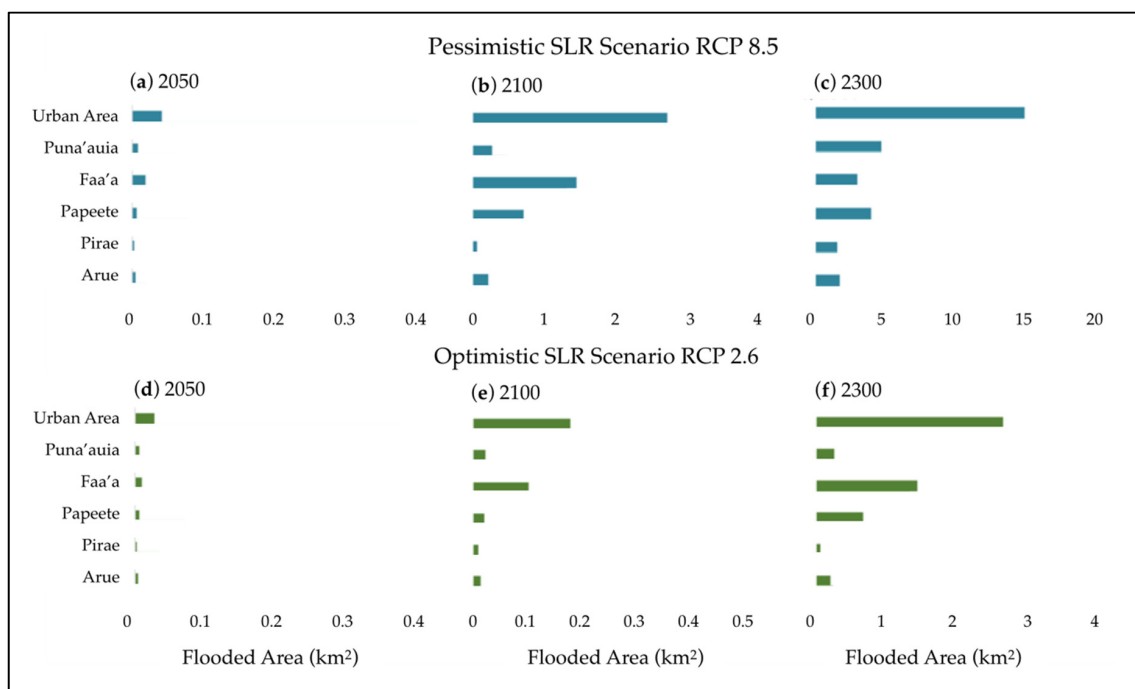

**Figure 5.** Comparison of flooded urban areas in Tahiti for pessimistic (**a**–**c**) and optimistic scenarios (**d**–**f**) from 2050–2300 [59].

In an optimistic scenario, virtually the same areas as in the pessimistic scenario are affected, but to a lower degree. Once again, the municipalities of Papeete and Faa'a are the most sensitive to SLR (Figure 4). In Papeete, 0.026% of the commune was observed to be flooded, or 0.0046 km$^2$ (0.46 hectares), while in Faa'a, 0.025% is flooded, or 0.0091 km$^2$ (0.91 hectares) [59]. Arue, Punaauia, and Pirae are at 0.020%, 0.006% and 0.004%, respectively.

2100:

In a pessimistic scenario, the newly flooded lands between 2050 and 2100 for the entirety of the urban area represents 1.47%, for a total of 2.74 km$^2$. In Faa'a, more than 1.46 km$^2$ of land is flooded, or 4.04%, while in Papeete, 0.72 km$^2$, or 4.09% is flooded (Figure 5). The impacts are also evident in the municipalities of Arue and Punaauia, where the area is reduced by 1.06% and 0.35%, respectively. These floods are mainly seen on the banks and outlets of the rivers (Figure 4). Pirae comes out with the lowest at 0.19%.

In an optimistic scenario, it is again Faa'a where most of the flooded land is observed, with a total of 0.1 km$^2$, or 0.29%. The impacts are mainly concentrated on the inner dyke of the airport runway and on the façade facing the sea (Figure 4). Flooding is particularly low in Punaauia and Pirae, where it does not exceed 0.03%. Cumulatively in the urban area, there is a net loss of only 0.18 km$^2$ (17 hectares), or 0.01% of its total area (Figure 5) [59]. Papeete and Arue are at 0.11% and 0.06%, respectively.

2300:

For this reference year, the municipalities are now all heavily affected in the pessimistic scenario. However, differing from the previous reference years, the commune of Faa'a suffers less impact between 2100 and 2300 compared to the other municipalities. It is in Papeete where the newest flooded lands are observed, 18.07%, for a total of 22.16% of its territory, between today and 2300 (Figure 5). Arue, Faaa, Punaauia, and Pirae are relatively lower at 8.22%, 8.13%, 5.95%, and 4.59%, respectively.

In an optimistic scenario, Faa'a is flooded over the first few hundred meters of its coastline and the airport is almost completely underwater. Thus, more than 1.40 km$^2$ is flooded, representing a proportion of 3.87% of the territory of the municipality. As for Papeete, 0.66 km$^2$ of its territory is flooded, for a total of nearly 3.75%. Although no municipality will be spared between now and 2300, Arue, Pirae and Punaauia will have the least impacts, with a 0.96%, 0.17% and 0.32% loss of territory, respectively.

In conclusion, in both scenarios, river mouths and beaches are the most impacted. The results for all three years, in all scenarios (Figure 5), show that flooding occurs much farther inland when there's a river or valley trough. Some rivers have been diverted or buried with the urbanization of the coastlines, but will be submerged by the rising sea. The results also show that in almost every combination of factors (scenario and/or reference year), it is the municipalities of Faa'a and Papeete that are most affected by SLR in the next 280 years [59].

### 3.3.2. Simulations for Impacted Infrastructure

2050:

Considering that the impacts of SLR for this year are extremely concentrated on the coastlines and river mouths, few buildings are affected in the RCP 8.5 scenario (Figure 6). Throughout the urban area, there are a total of 27 affected buildings, 21 of which are residential. Of these 27 buildings, eight are located in Papeete and eight in Faa'a (the two municipalities most affected by 2050), while there are also eight located in Punaauia; however, this area is less affected by floods (Figure 7). In Faa'a, all the affected housing is located on the banks of the river's drainage channel for major floods. Arue and Pirae have one and two buildings affected, respectively.

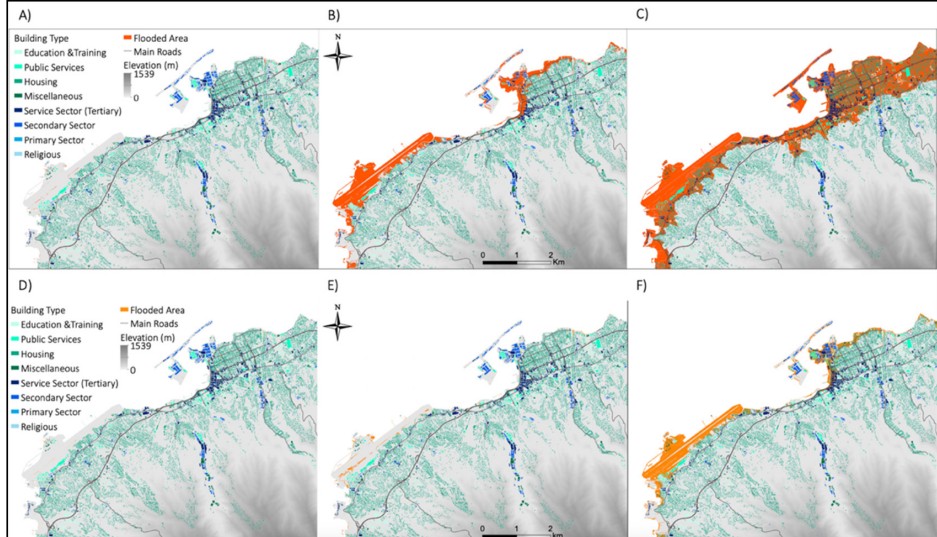

**Figure 6.** Infrastructures affected by SLR in the most affected portions of the municipalities according

to two scenarios of GHG emissions. (**A**) Pessimistic scenario in 2050. (**B**) Pessimistic scenario in 2100. (**C**) Pessimistic scenario in 2300. (**D**) Optimistic scenario in 2050. (**E**) Optimistic scenario in 2100. (**F**) Optimistic scenario in 2300 [59].

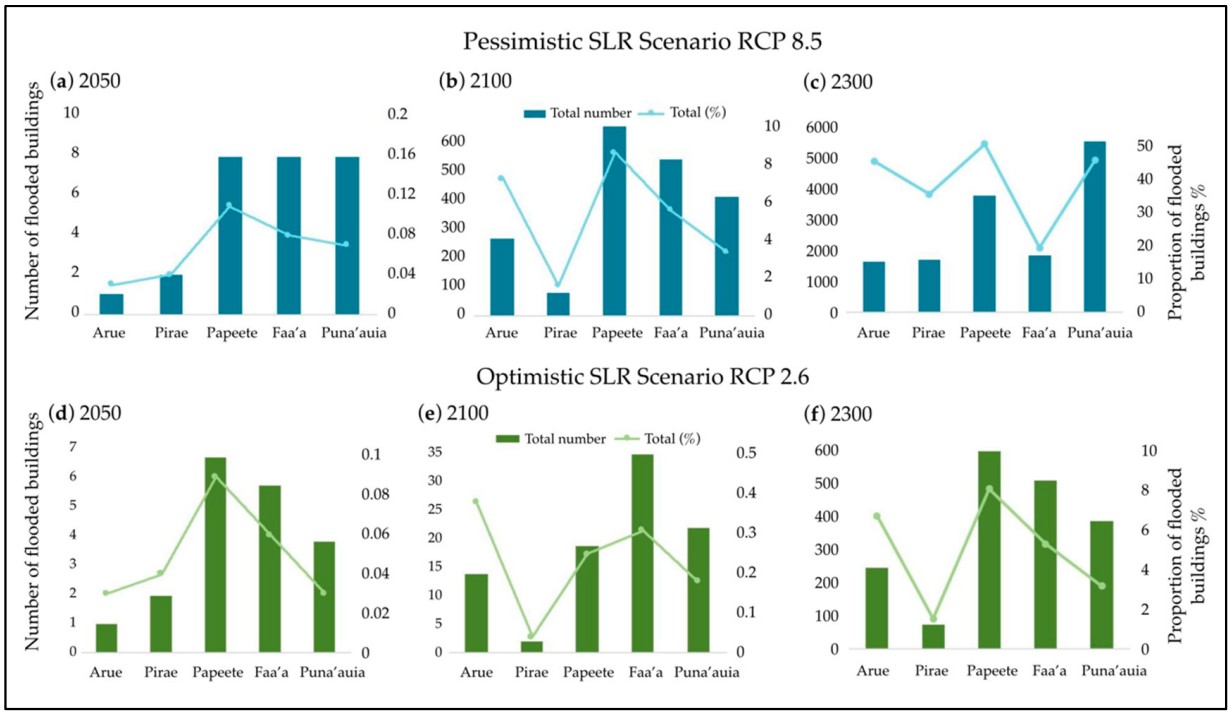

**Figure 7.** Number of flooded buildings and the proportion of the total amount of building in each municipality according to the pessimistic (**a–c**) and optimistic scenarios (**d–f**) of SLR values and the three years of references [59].

In the optimistic scenario, only 20 buildings are affected in the entire urban area by 2050, 14 of which are residential and 4 of which are service buildings (Figure 7). The majority of the housing buildings are in the municipalities of Faa'a and Papeete, while the service buildings are in the municipalities of Punauuia, Faa'a and Papeete [59]. Once again, Arue and Pirae have one and two buildings affected, respectively, with the remaining 17 dispersed mostly between Faa'a and Papeete (6 and 7).

2100:

Pessimistic scenario: there is a drastic increase in the number of buildings affected compared to 2050 (Figure 7). There are now 1939 flooded buildings, a proportion of more than 5% of all buildings in the urban area. The greatest number of affected infrastructures is in Faa'a (537 buildings) and Papeete (650 buildings), followed by Punaauia (410 buildings). Papeete is the municipality that loses the highest proportion of buildings (Figure 7), 8.62%, followed by Arue, where nearly 7.28% of all buildings are impacted. In the entire urban area, after housing, it is public service, tertiary service and secondary sector buildings that are most affected, with at least 14% impacted for these types of buildings. Pirae is the least impacted with 77 buildings affected, of which 93% are used for housing.

Since the values of sea level rise between 2050 and 2100 are relatively close in an optimistic scenario, the results are similar, as the majority of flooded buildings are either on river banks or very close to the sea. There are 71 new buildings affected in 2100 throughout the urban area, 86% of which are houses. The remaining buildings are, in descending order, secondary sector, miscellaneous and professional (tertiary) service buildings. Faa'a has the highest number of newly flooded buildings, with a total of 30. Surprisingly, when looking at the proportions of buildings affected, Arue (13 buildings) comes almost even with Faa'a,

with a difference of 0.01% [59]. Punaauia has 18 new buildings affected but with a relatively low percentage, while Pirae has zero new building impacted by the sea level rise.

2300:

The number of newly affected buildings explodes between 2100 and 2300 in the pessimistic scenario (Figure 7), with 12,561 new floods for a total of 14,500 buildings throughout the urban area, which is equivalent to 38.31% of all buildings. Papeete has the highest proportion of affected buildings, with nearly 50%, for a total of 3773 buildings, while Punaauia has the highest number of affected buildings, with a total of 5549, or 45.24%. Arue follows Punaauia very closely in terms of cumulative proportion with a total of 44.93% of its buildings flooded by 2300. In contrast to the other reference years, Pirae is finally impacted, with a cumulative total of 35.13% of buildings affected for a total of 1623 new buildings flooded or 4839 total buildings from 2050 to 2300. Faa'a has the lowest proportion of newly affected infrastructure (between 2100 and 2300) and cumulatively (for 280 years).

In comparison, in the optimistic scenario (Figure 7), only 4.83% of the buildings are now flooded, in the entire urban area, representing 1828 buildings (Figure 7). The municipality of Papeete has the highest proportion of impacted buildings (Figure 7) with 8.11%, followed by Arue with 6.73% and Faa'a with 5.31%. On the other hand, in terms of the number of buildings, the order is somewhat different: in decreasing order, the municipalities of Papeete (612 buildings), Faa'a (511 buildings) and Punaauia (389 buildings) are the most affected by SLR. In all municipalities, except Papeete, more than 80% of the buildings affected are residential. In Papeete, it is noted that nearly 30% of the buildings affected by 2300 are professional buildings in the secondary sector. In Faa'a, 18% of all service (tertiary) buildings would be flooded before 2300. In Arue, it is the public service buildings that would be the most affected with the flooding of nearly 25% of them [59]. Pirae is relatively untouched with a cumulative total of 1.51% from 2050 to 2300.

While these results remain modeled predictions of future scenarios, their validity cannot be blindly accepted due to the uncertainty of such events and parameters. However, it is important to note that even in the optimistic scenario for the year 2050 (only ~20 years from today), there are indications of submerged zones along the airport zone in Faa'a and the northern coast of Papeete (the capital city). This airport is the only international airport within the entirety of French Polynesia and serves as a primary import/export zone for the territory. The impacted zones therefore require additional data in order to continue assessing their vulnerability. Generally speaking, the number of infrastructures affected is relatively proportional to the extent of the impacts of the floods on the coasts, i.e., the municipalities with the most flooding (Faa'a and Papeete) are also those where the infrastructures are the most impacted. The flooding results show that for the two GHG emission scenarios and the three reference years, the municipalities mainly affected are Papeete and Faa'a. Since these two municipalities are the cultural, social and economic centers of Tahiti, the consequences of sea level rise would be even more damaging for the entirety of this island [59]. Although Papeete and Faa'a are the two most affected municipalities, in general, for the three reference years, we note that in 2300, in the pessimistic scenario, Faa'a has the least infrastructure loss. This could be explained by its very steep topography inland, beyond the airport grounds. Thus, even if the water level increases significantly, the slope at this location is so steep that it would serve as protection for a large number of infrastructures located higher up [59].

The results presented in this study demonstrate detailed scenarios of the impacts that sea level rise may have on the coastlines and infrastructure of the Tahitian urban area when taking into account the parameters of GHGs from the IPCC scenarios as well as the Antarctic factor from Kopp et al., 2017. The GHG emission scenarios were addressed to establish a window of opportunity for impacts; however, these monitoring efforts need to be conducted at a high temporal frequency in order to be as accurate as possible. Airborne LiDAR data produce high-precision digital models on a spatial scale that cannot be compared to other current aerial sources, and for that reason, this aerial source is an effective tool for coastal monitoring, globally and in French Polynesia. Despite its various advantages, Airborne

LiDAR remains very costly due to the price of the laser and associated mounting equipment, the plane, and an experienced pilot (the airplane has to fly low and almost perpendicular to the ground, which requires specific piloting skills) [72]. These factors call for massively high costs rendering the possibility of repetitive surveying improbable. Studies requiring daily, weekly, monthly, or even yearly surveys, cannot sustainably use airborne LiDAR data. Therefore, monitoring coastal environments and assessing coastal risks cannot be carried out only with this type of data. Repetitive monitoring at high temporal frequencies is a necessity for obtaining updated information to be able to inform decisions making entities.

## 4. Perspectives on Acquiring New Data, a Key to Developing More Precise and Localized Knowledge on Climate Risks in French Polynesia (Observatory Task 2)

### 4.1. UAVs for Coastal Management

Unmanned Aerial Vehicles (UAVs) offer a solution to these issues allowing for higher frequencies of monitoring campaigns at a relatively lower cost while obtaining similar levels of accuracy to LiDAR data, which is the most commonly used method today for the acquisition of aerial data [63]. Recent developments and improvements of UAVs have allowed UAVs to emerge as an important tool in environmental conservation and monitoring [61]. It is important to note that there exist a variety of different types of UAVs, from flying wings to multi-copter and ultra-light to large industrial UAVs, each with their own benefits and limitations. In addition to the wide variety of UAV types, UAVs are also multifaceted when it comes to their capabilities. These vehicles have the capacity to carry cameras, lasers, sensors, or even test tubes to acquire data [73]. UAVs serve as a mobile platform ultimately accompanying or even replacing other aerial data sources such as satellites or aircrafts, which require heavy manpower and specific technical knowledge. Studies range from attaching multispectral sensors to identify chlorophyll-a levels in bodies of water [73], to ordinary RGB cameras to obtain 3D models [59], to even attaching LiDAR sensors for the acquisition of digital surface models [74].

UAVs are capable of flying at high altitudes of 500 m but most importantly at lower altitudes (<100 m), which allows for data to be captured for a finer spatial resolution in outputs. Despite the fact that UAVs cannot cover a large surface area at one time due to their restricted flight time (generally < 1 h), they do have high levels of maneuverability, which increases spatial coverage and access to areas, previously unattainable by airplanes or helicopters such as dense forests or under the canopy. In the context of risk assessment, UAV-based monitoring of disaster events has significant advantages due to its rapid availability (low time mission planning) and high level of automation [62]. In addition, the possibility of immediately viewing the photographs in the field allows for repetition in case of errors. Other advantages include very low security risks in case of accidents, due to the light weight of these devices, and risk awareness capacities. UAV footage and imagery can also be used as a communication tool for community awareness to highlight various environmental hazards and engage stakeholders at various levels. Evidently, every data source comes with its limitations, and UAVs are no exception. Most multi-rotor UAVs have low flight times due to their weight and their batteries' capacities. Fixed-wing UAVs, on the other hand, can fly for more than an hour and a half. Additional limitations include: susceptibility to bad weather (rain, snow, sleet, high winds, etc), laws and regulations of UAV flight (specifically in France), lacking development of small light weight sensors and cameras, which are all examples of the shortcomings that can be encountered with UAVs [75]. However, the linking of this aerial data source with other current sources, and the technical advancements made over time, lead to UAVs are earning their spot in the field of coastal risk assessment.

In conclusion, UAV imagery is efficient in terms of cost, resolution, and acquisition time, rendering this method ideal for conducting repetitive surveys. However, this approach is still limited spatially and cannot be used to monitor large surfaces at a time. UAVs and their associated equipment produce data that are crucial for risk assessment and coastal monitoring. The ability of UAVs to perform missions and acquire data autonomously as

well as its maneuverability helps with complex coastal environmental disaster monitoring and has the potential to revolutionize the availability of data for spatial modelling. The technical advantages, high temporal frequency, and capacity for communication and community awareness tools argue that UAVs are an efficient tool, full of potential for data gathering to arm decision makers and local authorities with a more accurate picture of environmental impacts.

### 4.2. Examples of UAVs for Coastal Management and Application in French Polynesia

Remote sensing, stereo mapping, photogrammetry techniques, i.e., structure from motion, species surveys, and even bathymetric surveying (in clear waters) are all possible techniques (non-exhaustive list) for UAVs to provide data for coastal risk assessment and management. References to the use of UAVs for coastal monitoring in the scientific literature is relatively new but has been evaluated in several studies.

Photogrammetry techniques have become very popular when using UAVs in this domain because only an RGB camera is required, thus reducing the costs and materials needed to obtain high-precision models. The combination of modern UAVs and photogrammetry processes allows for techniques such as structure from motion to create 3D models from 2D images. In addition, through the photogrammetry software, UAV images are capable of being ortho-rectified, rendering non-distorted images which can be geo-referenced together to create orthomosaics. These digital models and orthomosaics are essential when attempting to assess and predict coastal risks [59–63,76–81]. Other studies have attached multispectral and hyperspectral sensors, thermal cameras, and even LiDAR lasers to UAV platforms; however, these attachments are much more expensive. Multi/hyperspectral sensors provide greater spectral information than regular RGB cameras and greater precision. Other sensors such as LiDAR have been identified as a viable solution for penetrating areas with high vegetation and still obtaining precise digital models. Attaching these sensors to UAVs rather than piloted aircrafts increases maneuverability and access to complex terrain, as well as decreasing prices and time required for a flight mission [73,82–88]. The feasibility of UAVs for coastal management is thus attested for.

The use of UAVs for coastal zone management (CZM) is unprecedented in Tahiti, with only a handful of other studies in the entirety of French Polynesia. These studies use UAVs for marine species monitoring [80,89,90] and remote sensing as well as photogrammetry techniques for coral reef mapping [77,91]. This spatial tool has great potential in the context of coastal risk analysis in French Polynesia and the utilization and development of this tool is important for the management of the increasingly vulnerable coasts of this island nation.

### 5. Discussion

This article describes two tasks of risk and resilience observation in French Polynesia. This spatial decision support system aims to support the implementation of resilience on a long-term and local scale. For this purpose, the observation is divided into six tasks, all of which are essential for the proper functioning and sustainability of the tool. The tasks developed in this paper illustrate the need for data in order to increase knowledge of risks and to implement adequate resilience strategies.

An initial research effort was executed to supply Task 1, modeling the impact of sea level rise within the context of climate change on the territory of the Tahitian agglomeration as well as its critical infrastructures. This study made it possible to utilize data previously acquired and used by Polynesian actors (LIDAR data), but also to combine a pre-existing methodology in order to integrate data with regard to the Antarctic melting factor. This approach allowed for the mapping of the impacts of the sea level rise on the agglomeration as well as specifying the analysis by integrating the impact on critical infrastructures. This study is all the more relevant today, as the Pacific island territories are rarely taken into account in the official projections [10]. Indeed, the modeling of climate change scenarios has difficulty integrating these territories, particularly because of the predominance of the sea

over land, which greatly distorts the projections. Global models are therefore not adequate for a spatialized and contextualized decision support system for French Polynesia.

Despite showing its necessity in the process of acquiring knowledge and its long-term integration, this study is limited by the use of LiDAR data which are excessively costly and complex to perform repetitively for updated measures. Using locally produced data and processing them with the perspective of analyzing the impacts of climate change favors an adoption by local actors. However, it is necessary to point out the limitations of this data and to look toward the future where data acquisition must be more systematic, less costly (specifically in the case of Pacific islands), and sustainably effective over time. Therefore, based on the limitations of task 1 (Table 4) of the observatory, it was necessary to expand the data acquisition task. There must be a steady inflow of data coming from repetitive surveys in order to accurately assess the current state of the coastal zones being examined [92]. As stated previously, other aerial data sources, (while they have their own advantages) do not allow for surveys at high temporal frequencies. UAVs are becoming a necessity to be able to sustainably assess coastal zones. Combining different aerial data sources such as satellites, airplane photographs, LiDAR, and UAVs will provide different yet valuable information for risk assessment. Together, these sources are capable of providing the raw data for effective coastal management [63]. The potential of UAVs has been defined by showing it is the only source of aerial data capable of performing repetitive surveys. This specific aerial source requires further efforts for application as it is relatively unprecedented in the context of French Polynesia in order to continue solidifying the resilience observatory.

**Table 4.** Main advantages and limits in the data acquisition methodologies.

| | Pre-Existing Data (LIDAR) | New Data (UAVs) |
|---|---|---|
| Advantages | - Immediately accessible<br>- Large spatial coverage<br>- High resolution | - Low cost<br>- Repetitive surveys<br>- ustainable over time<br>- High resolution<br>- High acquisition time<br>- Multiple sensor attachment<br>- Capacity for obtaining own data |
| Limits | - Very expensive<br>- Cannot be reproduced over time<br>- Dependent on large acquisition campaigns | - Requires flying skills and expertise<br>- Limited spatial coverage<br>- Laws and regulations |

## 6. Conclusions

In the context of climate change and associated uncertainties, developing decision-support tools has become vital. For a variety of territories, their vulnerability extends to the increasing risks as well as the lack of tools to implement more resilient risk management strategies. The Pacific Island territories, particularly French Polynesia, are among the most at-risk territories, hence the development of the risk and resilience observatory project in order to overcome the lack of knowledge, tools, data and resources that French Polynesia is currently facing. This observatory is composed of six bricks to ensure a sustainable implementation in the territory. Two of these bricks have been exemplified in this article, with Task 1: increasing the knowledge of risks and Task 2: acquiring and modeling data. These two tasks were supported by specific studies. The first aimed to model the impact of sea level rise on the Tahitian territory and local critical infrastructures with pre-existing data. The objective was to enhance and exploit pre-existing data, beyond their limitations, to produce information and allow projections of the impact of sea level rise over the long term (2050–2300). The second study serves as a progress report aimed at addressing the limitations of pre-existing local data as well as proposing a new type of data acquisition, more appropriate to the local territory, the application of UAVs.

The acquisition, production, processing and analysis of pre-existing or new data is an important part of the conscious and spatialized decision support process. However, producing raw data without an analytical framework cannot be efficient over time. Therefore, it is essential to develop spatial decision support system in order to centralize the organization,

collection and processing of monitoring data in a comprehensive way [93]. For such a system to be effective, it must be catered to a specific environment in order to integrate the proper variables of the environment being observed. The observatory of resilience responds to this need of catering to the local environment and will serve as a spatial decision support system in French Polynesia. The resilience observatory is therefore the keystone of the valorization and production of data and provides an analytical framework and a support for knowledge transfer at the local level.

**Author Contributions:** Conceptualization, D.S.; methodology, C.H., D.S. and J.J.; software, J.J.; formal analysis, J.J.; writing—original draft preparation, C.H. and J.J.; writing—review and editing, C.H., D.S., J.J. and N.L.; supervision, D.S. and N.L.; project administration, D.S.; funding acquisition, D.S. All authors have read and agreed to the published version of the manuscript.

**Funding:** This research received no external funding.

**Data Availability Statement:** The raw topographic data and datasets analyzed in this study were provided by the French Polynesian Urban Planning Department (SAU) led by the Hydrographic and Oceanographic Service of the Navy (SHOM).

**Conflicts of Interest:** The authors declare no conflict of interest.

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
