# Peer review of "Supporting a Resilience Observatory to Climate Risks in French Polynesia: From Valorization of Preexisting Data to Low-Cost Data Acquisition"

_water, doi:10.3390/w14030359_

Round 1
Reviewer 1 Report
I quite agree with the development of decision support tools for climate change risk proposed in this study, but I believe the content presented in the manuscript is not sufficient for a research paper, and the logic, data, methods, content and conclusion of the paper need to be greatly improved to meet the needs of publication. Here are the specific comments:
- The logic of the introduction is not clear enough to explain the necessity and main objective of this study.
- Line43, 0,3? The same problem appears in Table 1.
- Starting with Table 2, the first rows and columns of the tables in the manuscript are missing.
- Line314, Description of data and methods is not clear enough.
- Section 4 is more of a progress review than a study.
Author Response
- The logic of the introduction is not clear enough to explain the necessity and main objective of this study.
- Edits were made to clear up the division of the 4 sections by adding a concluding paragraph at the end of the introduction section.
- Line43, 0,3? The same problem appears in Table 1.
- Formatting error fixed
- Starting with Table 2, the first rows and columns of the tables in the manuscript are missing.
- Formatting error fixed
- Line314, Description of data and methods is not clear enough.
- Missing paragraph of data and methods was added. Another formatting error.
- Section 4 is more of a progress review than a study.
- Changes made identifying this section as a progress review were noted in introduction of article when defining the objectives and in the conclusion. Additionally, Title of section 4 therefore altered to:
- Perspectives on acquiring new data, a key to developing more precise and localized knowledge on climate risks in French Polynesia (Observatory Task 2)
Reviewer 2 Report
The article is of a review rather than original character. This is evidenced by the fact that it contains almost 100 cited scientific publications. It's too much! Nevertheless, it has some advantages, as we obtain a summary of the results relating to climate change in this exotic area of ​​the Earth. The authors' own contribution concerns potential changes related to sea level rise. This may be the subject of publication. Although I approach this type of publication without enthusiasm.
I have two comments
# 46-48 "… .climbing temperatures which are increasing in recurrence and intensity of extreme events such as floods, tropical hurricanes and tsunamis due to climate change". Isn't rising temperature climate change?
493-494: "The results presented in this study demonstrate in detail the impacts that sea level 493 rise will have on the coastlines and infrastructure of the Tahitian urban area"
The authors are uncritical about the obtained results, which are some kind of evaluation. Will it surely be so. In my opinion, this is very doubtful. These are just scenarios. So it needs to be corrected.
Author Response
- 46-48 "… .climbing temperatures which are increasing in recurrence and intensity of extreme events such as floods, tropical hurricanes and tsunamis due to climate change". Isn't rising temperature climate change?
- This sentence we re-phrased to correct this statement. (see lines 46-48)
- 493-494: "The results presented in this study demonstrate in detail the impacts that sea level 493 rise will have on the coastlines and infrastructure of the Tahitian urban area." The authors are uncritical about the obtained results, which are some kind of evaluation. Will it surely be so. In my opinion, this is very doubtful. These are just scenarios. So it needs to be corrected.
- Paragraphs were added to clear up this statement. (see lines 575- 582 and 595-599)
- English revisions
- Document was revised by native English speaking colleague.
Round 2
Reviewer 1 Report
Table 1 has no units
Please unify the font and font size of the text.
The font in Figure 3 is suggested to be enlarged a bit.